# Nano-Graphitic based Non-Volatile Memories Fabricated by the Dynamic Spray-Gun Deposition Method

**DOI:** 10.3390/mi10020095

**Published:** 2019-01-29

**Authors:** Paolo Bondavalli, Marie Blandine Martin, Louiza Hamidouche, Alberto Montanaro, Aikaterini-Flora Trompeta, Costas A. Charitidis

**Affiliations:** 1Thales Research and Technology, 1 Av Augustin Fresnel, 91767 Palaiseau, France; marie-blandine.martin@thalesgroup.com (M.B.M.); hamidouche.louiza@gmail.com (L.H.); alberto.montanaro@cnit.it (A.M.); 2Research Unit of Advanced, Composite, Nanomaterials and Nanotechnology, School of Chemical Engineering, National Technical University of Athens, 9 Heroon Polytechneiou st., Zografos, GR-15773 Athens, Greece; ktrompeta@chemeng.ntua.gr (A.-F.T.); charitidis@chemeng.ntua.gr (C.A.C.)

**Keywords:** ReRAM, carbon nanofibers, spray-gun deposition

## Abstract

This paper deals with the fabrication of Resistive Random Access Memory (ReRAM) based on oxidized carbon nanofibers (CNFs). Stable suspensions of oxidized CNFs have been prepared in water and sprayed on an appropriate substrate, using the dynamic spray-gun deposition method, developed at Thales Research and Technology. This technique allows extremely uniform mats to be produced while heating the substrate at the boiling point of the solvent used for the suspensions. A thickness of around 150 nm of CNFs sandwiched between two metal layers (the metalized substrate and the top contacts) has been achieved, creating a Metal-Insulator-Metal (MIM) structure typical of ReRAM. After applying a bias, we were able to change the resistance of the oxidized layer between a low (LRS) and a high resistance state (HRS) in a completely reversible way. This is the first time that a scientific group has produced this kind of device using CNFs and these results pave the way for the further implementation of this kind of memory on flexible substrates.

## 1. Introduction

The dynamic spray-gun deposition method is a new, revolutionary technique that, thanks to its versatility, can be implemented for a large range of applications. In this context, an extremely promising application is the fabrication of low-cost graphite-based memories that can be integrated with flexible, plastic, or paper-based substrates. In the International Technology Roadmap for Semiconductors (ITRS) of 2011 [1], within the chapter concerning the Emerging Research Devices (ERDs) and, more specifically, memory devices, it was stated that ultrathin graphite layers are “interesting materials for macromolecular memories, thanks to the potential fabrication costs that are considered as the primary driver for this type of memory, while extreme scaling is de-emphasized”. Indeed, these memories can be implemented in low-cost applications and integrated, for example, in ID cards, driver licenses, smart-cards, and on paper, as well as potentially being patched for health applications; the implementation of these memories demonstrate the possibility of storing information and changing it in an easy way. Up until now, there has been no commercialized technology available for flexible, low-cost memories. In this piece, we will describe how Metal-Insulator-Metal (MIM) structures were fabricated, where the carbon-based materials were sandwiched between two metal contacts. This kind of structure is a Resistive Random Access Memory (ReRAM), in which the resistance of the sandwiched material is changed by applying a bias between the top and bottom contact.

The first work highlighting the utilization of graphene-based mats for flexible, non-volatile ReRAM was published in 2010 by Jeong et al. [2]. In this paper, by exploiting the hydrophilic character of Graphene Oxide (GO) flakes, they achieved stable suspensions and deposited them on large surfaces through a spin coating process. The work of Jeong et al. demonstrated the non-volatile effect on the resistance of a 70-nm-thick layer of GO flakes consisting of a layered structure of Al/GO/Al in a cross-bar configuration. Another interesting piece of work was produced by He et al. in 2009 [3], when reliable and reproducible resistive switching behaviors were observed in GO thin films prepared by the vacuum filtration method, a common technique used to fabricate Carbon Nanotube (CNT)-based bucky papers. The main hypothesis regarding the physical principle at the origin of the switching effect was surrounding the absorption/desorption of oxygen-related groups on the GO sheets, as well as the diffusion of the top electrodes. Indeed, the alignment of the oxygen vacancies creates conducting paths that reduce the resistance of the sandwiched layers moving from a high-resistance state (HRS) to a low-resistance state (LRS) [4]. Another hypothesis concerned the oxidation of the top contact. The pioneering work of Hong et al. [5] included a deep analysis of the switching mechanism of this kind of device, underlining that these structures’ performances depend on the origin of the top contact. For example, in the case of Au-based top electrodes, there was no oxygen migration in opposition to Al electrodes. This can be explained because Au contacts cannot be oxidized. To evaluate the effect of the oxidation, Hong et al. performed measurements by fabricating contacts with different dimensions. As suspected, if the change in the resistance is related to the formation of dendrites, it does not depend on the contact dimensions [6]. Hong and coworkers demonstrated that the current between the two contacts was a function of the dimensions of the top contact and was therefore proportional to the number of potential conductive patches built-up by the oxygen vacancies. All of these experiments underlined the fact that GO was potentially useful for future non-volatile memory applications.

According to the above, our work focused on oxidized carbon nanofibers (CNFs) together with GO, a combination that could overcome the challenges that were faced in the previous studies.

## 2. Materials and Methods

### 2.1. Definition of the CNFs’ Specifications

Taking into account the aforementioned preliminary results, this study aimed to fabricate memories based on oxidized CNFs [7]. CNFs are less expensive in comparison with CNTs and are easier to be fabricated. In this case, graphene worked only as the “vehicle” of the oxygen atoms, i.e., graphene was the carrier of the necessary oxygen [8]. Considering that the final thickness of the insulator layer needed to be around 50 nm, extremely diluted suspensions using oxidized CNFs in de-ionized water were used, as described in the following.

### 2.2. Carbon Nanofibers’ Growth, Purification, and Functionalization

CNFs were synthesized in the Research Unit of Advanced, Composite, Nanomaterials, and Nanotechnology, of the National Technical University of Athens, through the Thermal Catalytic Chemical Vapor Deposition technique (TC-CVD). The specifications of the synthesized materials were defined as described in 2.1, to fit the application. The TC-CVD method is a versatile technique, through which structures with different geometries can be produced [9]. Thus, a length of ~5 μm and a diameter of 50 ± 5 nm were aimed for, which was achieved with the use of the appropriate catalyst and was synthesized in-house. The catalyst consisted of Fe^3+^ particles, embedded on Al_2_O_3_, which had been prepared via the impregnation method [10]. The parameters of the CVD (Chemical Vapor Deposition) reaction also played a crucial role in the obtained structures. Thus, after a series of preliminary experiments, a temperature range of 700–750 °C was chosen for the reaction to take place, together with a total flow rate of 330 mL/min for the reaction gases (acetylene and nitrogen of a volume ratio of 1:2, respectively). 

After their synthesis, CNFs were purified to remove any amorphous by-product remaining, which could be observed as spherical agglomerates in a fluffy form. Moreover, the purification enabled the removal of the metallic catalyst residues. In order to purify the sample, a step-by-step chemical procedure took place. Initially, the sample was stirred in NaOH (1 mol/L) for 2 h at 80 °C to remove the Al_2_O_3_ substrate. Afterward, the dried sample was extracted with 5 mol/L of HCl overnight in a Soxhlet extractor at the boiling point of the acid. Washing took place with distilled water extensively, in order to remove the dissolved catalyst. The success of the purification was confirmed with Energy Dispersive X-Ray Spectroscopy (EDS) analysis, the results of which are presented below (Figure 1). As can be seen, Fe has been totally removed (0.0 wt.%), while Al from the Al_2_O_3_ substrate is up to 1.8 wt.%. The 5 wt.% Si refers to the Si wafer substrate, where the CNFs’ growth took place. In Figure 1, the morphology of the purified CNFs is also presented, which was studied through Scanning Electron Microscopy (SEM). The CNFs produced form entangled bundles with straight structures of less than 5 μm. Their surface is smooth and their diameters are 50 ± 5 μm. Figure 2 presents the CNFs’ internal structure, which has been studied through Transmission Electron Microscopy (TEM). The interconnections of the inner sidewalls are clearly visible. Their internal cavity is also able to be observed and these nanofibers are classified as being stacked-cup.

The final step prior to the use of the CNFs in the spray-gun deposition is their oxidation. Chemical functionalization of their surface was carried out with the treatment of strong acids [11]. Specifically, a mixture of 12 mol/L H_2_SO_4_/HNO_3_ in a volume ratio of 3:1 was used. The CNFs were boiled under reflux (~120 °C) in the acid mixture overnight. Then, intense washing with distilled water took place through several repetitions of centrifugation and a final step of vacuum filtration. By this point, oxidation of their surface was achieved, since the oxygen groups were anchored (–COOH, –OH, =O). After the described chemical functionalization, the CNFs were dispersed in de-ionized water and the stabilized suspensions were ready to be sprayed. Considering that the CNFs were oxidized, extremely stable suspensions were obtained. This enabled the development of a green-type process and a reduction in the heating temperature on the heating chuck; in the case of N-Methyl-Pyrrolidone (NMP, the most common solvent for carbon-based nanomaterials), the heating temperature is larger than 202 °C, but in the case of water, it is only 100 °C. 

### 2.3. Carbon Nanofiber-based Suspensions for the Spray-Gun Deposition Method

The suspensions were obtained using 10mg of oxidized CNFs in 500 mL of de-ionized water. Centrifugation was performed for 20 min, in two phases, at 3000 rpm. The top parts of the suspensions that were centrifuged was removed. This step is important in order to avoid the utilization of the bundles of high thickness; a critical parameter for the final fabrication of the memories. Indeed, the increased roughness can create dendrites (aligned metal particles linking the two electrodes) and, therefore, can lead to the short-circuiting of the device. The parts of the suspensions that were removed were further diluted in 500 mL of de-ionized water and sonicated in a bath for 6 h. Afterward, the deposition of a 2″ silicon-based substrate, that was preliminarily metalized using a Ti/Pt layer, was performed. The equipment used to spray the suspensions has been developed at Thales Research and Technology, using a patented process [12], and it constitutes the first prototype of this type of machine. This technique has already been implemented for other applications, such as gas sensing [13,14,15,16] and energy storage [17,18,19]. A picture of the machine set-up is shown in Figure 3 [18,20]. To perform the spray, the nozzle can move in two directions (x and y, see Figure 3), while the z-direction is set-up at the beginning. Due to this, samples with dimensions of 15 cm × 15 cm can be sprayed. The substrate is positioned on a heated chuck that can reach 250 °C. The temperature adjustment is an important parameter; in order to avoid the “coffee ring effect”, the drops reaching the substrate are heated at the boiling point of the solvent, which is 100 °C in this case, with water. Due to this, the nanomaterials in the drops are stuck in the impact zone and cannot move to the borders of the drop, achieving a uniform deposition.

In the following pictures, the top view of the sample is shown at different time points of the deposition of the suspensions. The last picture corresponds to one liter of the suspension having been deposited. We can see that, in Figure 4f, the couverture of the sample is optimized. The final thickness is around 150 nm. 

On top of the deposited CNFs layer, we patterned 100 μm × 100 μm squares where we then deposited AlCu/Au contacts. The active memory stack is depicted in Figure 5: 

## 3. Results and Discussion

To investigate the change in the resistance of the sandwiched CNF samples, an I-V (current-voltage) measurement was made using a probe station and a Keithly SMU module. The samples were placed on the chuck of the probe station, with two probes being used, one to contact the top electrode and the other to contact the back one. A bias was applied and the current flowing through the CNFs was measured. The sweep of the bias was made firstly from 0 V to −1.5 V, then from −1.5 V to 1.5 V, before moving back from 1.5 V to 0 V again. 

One hundred cycles like the one described were performed. The measured current for one cycle is shown in Figure 6. The oxidized CNFs are in contact with the inert bottom electrode of Pt and the easily-oxidized top electrode of AlCu. By applying a positive voltage between the bottom and top electrode, the oxygen atoms migrate from the oxidized CNF layer to the AlCu electrode. By applying the opposite voltage, the oxidized AlCu electrode gives back the oxygen atoms to the CNF-based layer. This back and forth movement of the oxygen atoms leads to a hysteretic change. 

One must note that the purity of the CNFs is essential to obtain this kind of behavior. In the case of non-purified CNFs, pinning of the memory has been observed, meaning that after a few cycles, the memory stays stuck in one state. This issue has been overcome by using purified CNFs, in which the memory stands show typical hysteresis cycles without being trapped in one state, as depicted in Figure 7. The tests showed that the memory behavior was retained throughout the 100 cycles.

If the principle of the memory has been understood correctly, many aspects are still to be investigated, such as the number of cycles the memory can withstand and the aging of the memory. By carefully observing Figure 7, it is clear that we have to understand why a sort of instability of the cycles exists, causing them to have the tendency to shift, even if their on/off ratio is maintained. However, their easy fabrication and their high rate of success, in terms of cycles observed for each memory row, make us believe that this new type of memory is very encouraging. 

## 4. Conclusions

This paper describes, for the first time, the fabrication of memories based on oxidized CNFs. This is the first time that a scientific group has demonstrated the possibility of producing memories using these materials. Moreover, these Resistive Random Access Memories (ReRAMs) have been fabricated using a completely new technique based on the dynamic spray-gun deposition method, patented by Thales. The importance of these results has to be taken into account because the CNF memories obtained on hard substrates can potentially pave the way for the revolution of memories on flexible substrates which, at the moment, do not exist as a commercial component in the market. Due to this, we would be able to widen the implementation of the technique to large market applications, such as ID cards, smart cards, memories for health monitoring on specific patches (e.g., diabetes monitoring), and ticketing. CNF-based memories can be fabricated roll-to-roll using the spray-gun deposition method, thereby dramatically reducing the cost of the devices. They can also be fabricated on paper-based substrates, creating disposable components, or integrated into radio-frequency identification (RFID) on plastics to add to the capability of storing information. However, a lot of work has to be done. Indeed, we have to understand why the cycles are unstable; the on/off ratio for each cycle is nearly the same but the current is increased. In addition, the metal contact appears to have a very important influence, considering that if we perform more measurements on the same byte, we do not obtain exactly the same graphs, even if the overall hysteresis behavior is always present. The other main problem is the identification of the exact formatting bias that activates the hysteresis cycle. For this reason, we need to do more systematic measurements on more samples, even if the phenomenon identified is very impressive considering the equipment used to achieve these kinds of structures and their potential implementation on a roll-to-roll production pilot line. 

## Figures and Tables

**Figure 1 micromachines-10-00095-f001:**
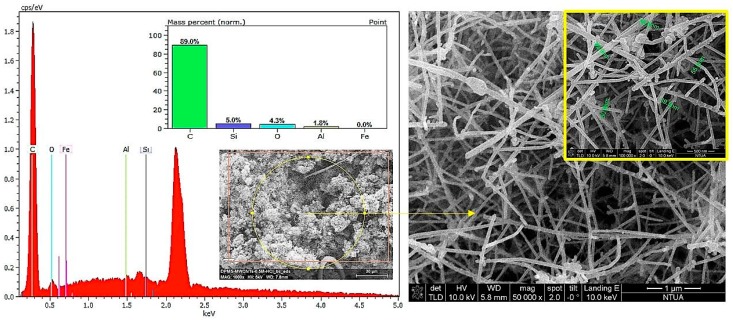
SEM images of the produced carbon nanofibers (CNFs) in (50,000× and 100,000× magnification) and an Energy Dispersive X-Ray Spectroscopy (EDS) analysis of CNFs for the purification assessment.

**Figure 2 micromachines-10-00095-f002:**
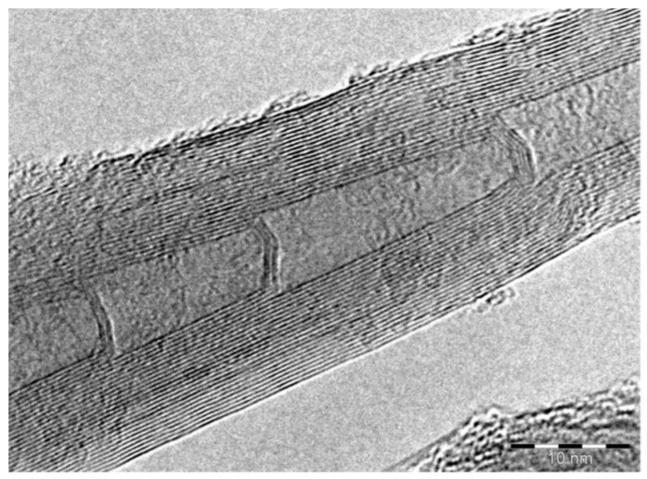
A Transmission Electron Microscopy (TEM) image of the produced carbon nanofibers (CNFs), showing the interactions of the inner sidewalls.

**Figure 3 micromachines-10-00095-f003:**
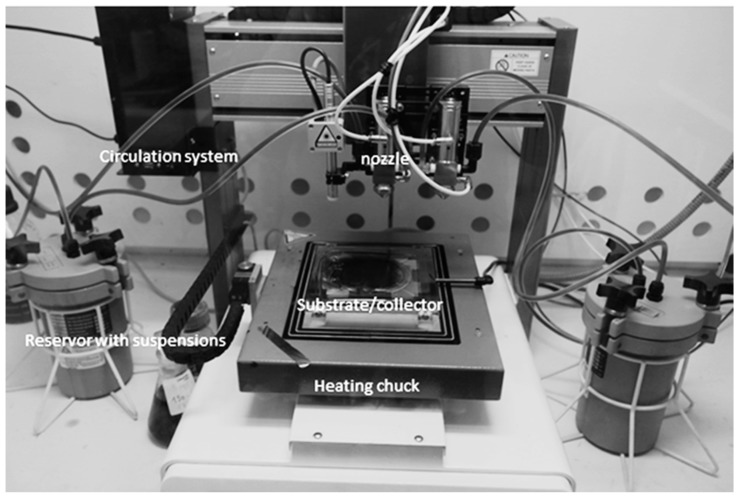
The dynamic spray-gun set-up developed at Thales Research and Technology.

**Figure 4 micromachines-10-00095-f004:**
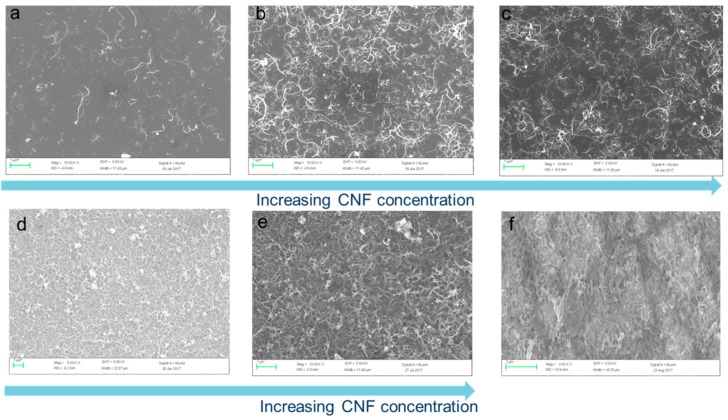
SEM images of carbon nanofibers (CNFs) deposited with the spray technique. The coverage rate increases until full coverage is reached in Figure 5e (single nozzle) and 5f (double nozzle). The concentration of the CNFs is increased due to the deposition time (**a**) 3 μm^−1^ (**b**) 6 μm^−1^ (**c**) 10 μm^−1^ (**d**) 15 μm^−1^ (**e**) 30 μm^−1^ (**f**) 60 μm^−1^ (the number corresponds to the CNFs, but an error of around 20% must be taken into account).

**Figure 5 micromachines-10-00095-f005:**
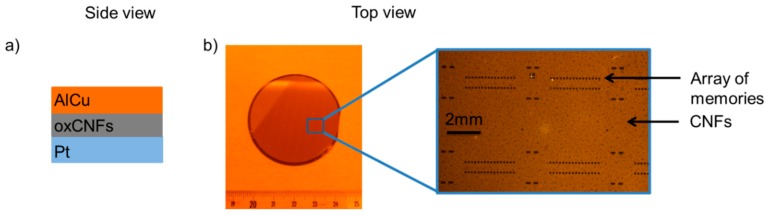
The side and top views of the sample.

**Figure 6 micromachines-10-00095-f006:**
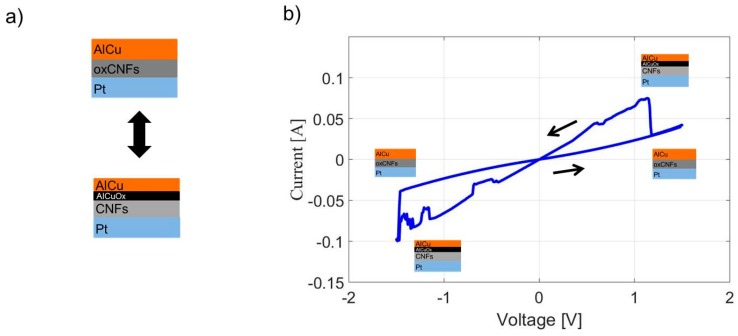
(**a**) The schematic side view of the two states of resistance of the memory stack (**b**) An image of a cycle obtained by applying a voltage between the inert Pt electrode and the AlCu electrode. The hysteretic change of current results from the migration of oxygen atoms between the carbon nanofiber (CNF) layer and the AlCu metal.

**Figure 7 micromachines-10-00095-f007:**
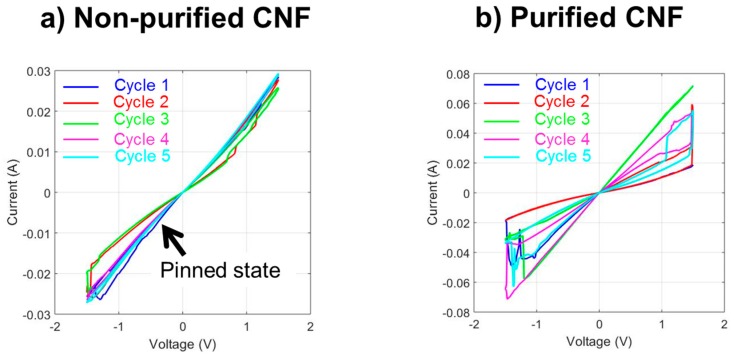
(**a**) Non-purified carbon nanofibers (CNFs) lead to memories being pinned in one state after a few cycles. (**b**) A memory fabricated with purified CNFs withstands many cycles without being pinned in one state.

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
