# Peer review of "Nano-Graphitic based Non-Volatile Memories Fabricated by the Dynamic Spray-Gun Deposition Method"

_micromachines, 2019, doi:10.3390/mi10020095_

Round 1

Reviewer 1 Report

This is an interesting paper, describing a new way to make low-cost memories based on oxidized graphene deposited by spray coating. 

The main concern about this paper is Figure 9b. Indeed, the latter apparently shows a few cycles of the developed ReRAM memory. However, it is not possible to understand which cycles they are. It is fundamental that the authors explain which cycles are described (1st, 2nd, 3rd and so on). Here one could think that memory aging occurs very quickly and that a stuck state is reached after a few cycles only. Please be more specific in the description of this figure. 

Here are minor comments.

Page 6 line 156, give the number of cycles performed."Many" is too vague. 

Figure 7 caption, use top/bottom rather than left/right.

Figure 6. It would be better to add the substrate on the side view. 

Figure 5. What does exactly mean "increasing CNF concentration"? From which value to which one ?

Page 4 line 125. The authors should cite the figures numbers. 

Page 4 line 116. What does t/m mean?

Page 2 line 84. It is Figure 2, not 3.

Author Response

Dear Reviewer,

Thank you very much for the very constructive remarks.

We answered all the requirements.

Thank you again.

Best Regards

The main concern about this paper is Figure 9b. Indeed, the latter apparently shows a few cycles of the developed ReRAM memory. However, it is not possible to understand which cycles they are. It is fundamental that the authors explain which cycles are described (1st, 2nd, 3rd and so on). Here one could think that memory aging occurs very quickly and that a stuck state is reached after a few cycles only. Please be more specific in the description of this figure.

Done. Added all the requested details.

Here are minor comments.

Page 6 line 156, give the number of cycles performed. "Many" is too vague.

Done. Changed with “one hundred”.

Figure 7 caption, use top/bottom rather than left/right.

Done.

Figure 6. It would be better to add the substrate on the side view.

The thicknesses are so small that it is difficult to perform a picture of the cross-section. We are dealing with nm.

Figure 5. What does exactly mean "increasing CNF concentration"? From which value to which one ?

Indeed it is more a question of deposition time that of concentration in the suspensions. We increased the deposition time in order to cover completely the substrate. The concentration of the suspension is provided. In the last sample we sprayed one liter.

Page 4 line 125. The authors should cite the figures numbers.

Added “fig.4”

Page 4 line 116. What does t/m mean?

Tours/minute, I changed with RPM rounds for min

Page 2 line 84. It is Figure 2, not 3.

Done

Reviewer 2 Report

This paper entitle “Nano-Graphitic based Non-volatile memories fabricated by dynamic spray-gun deposition method” provides a novel fabrication method of economic nano-graphitic ReRAM memories based on oxidized carbon nanofibers. However, this work lacks of many important errors which should be addressed in order to be suitable for publication.

Comment 1:

The acronyms ReRAM and MIM should be defined in the abstract and ITRS, ID in the introduction section.

Comment 2:

The introduction section should be enriched to add more references related to the applications of carbon nanofibers and fabrication methods of memories.

Thus, for example, they authors should mention other important applications of carbon nanofibers in the composite material field [Polymers, 10, 405 (2018). DOI: 0.3390/polym10040405 ] which produces lower cost composite materials than those produced with other carbon nanomaterials such as graphene oxide [Scientific Reports 7:11684 (2017) DOI:10.1038/s41598-017-10260-x (2017)]

Comment 3:

The English language should be revised by a native speaker because there are many mistakes such as those present in the following sentences:

-       The dynamic spray-gun deposition method is new revolutionary technique that thanks to its versatility can be implemented for a large panel of applications. (lines 26 and 27)

-       Indeed, in the ITRS roadmap of 2011 [1], it was stated in the chapter concerning the Emerging Research Devices (i.e. ERD) and more specifically on memory devices, that ultrathin graphite layers were “interesting materials for macromolecular memories, thanks to the potential fabrication costs that are considered as the primary driver for this type of memory, while extreme scaling is de-emphasized”. (lines 29-33)

-       Etc.

Therefore, the English language should be revised carefully.

Comment 4:

Many sentences in the introduction section should be referenced (lines 57-65).

Comment 5:

In line 84, Figure 3 probably means Figure 2.

Comment 6:

Sections 1 and 2 are mixed up (introduction, materials and methods and results and discussion) and then section 3 deals with results again

Comment 7:

Some reference such as reference 1 should be written according to the journal requirements

Author Response

Dear Reviewer,

thank you very much for the remarks.

Aswers embedded.

Thank you very much

Best Regards

This paper entitle “Nano-Graphitic based Non-volatile memories fabricated by dynamic spray-gun deposition method” provides a novel fabrication method of economic nano-graphitic ReRAM memories based on oxidized carbon nanofibers. However, this work lacks of many important errors which should be addressed in order to be suitable for publication.

Comment 1:

The acronyms ReRAM and MIM should be defined in the abstract and ITRS, ID in the introduction section.

Done.

Comment 2:

The introduction section should be enriched to add more references related to the applications of carbon nanofibers and fabrication methods of memories.

Thus, for example, they authors should mention other important applications of carbon nanofibers in the composite material field [Polymers, 10, 405 (2018). DOI: 0.3390/polym10040405 ] which produces lower cost composite materials than those produced with other carbon nanomaterials such as graphene oxide [Scientific Reports 7:11684 (2017) DOI:10.1038/s41598-017-10260-x (2017)]

I added these refs.

Comment 3:

The English language should be revised by a native speaker because there are many mistakes such as those present in the following sentences:

-       The dynamic spray-gun deposition method is new revolutionary technique that thanks to its versatility can be implemented for a large panel of applications. (lines 26 and 27)

-       Indeed, in the ITRS roadmap of 2011 [1], it was stated in the chapter concerning the Emerging Research Devices (i.e. ERD) and more specifically on memory devices, that ultrathin graphite layers were “interesting materials for macromolecular memories, thanks to the potential fabrication costs that are considered as the primary driver for this type of memory, while extreme scaling is de-emphasized”. (lines 29-33)

-       Etc.

Therefore, the English language should be revised carefully.

 We checked all the sentences.

Comment 4:

Many sentences in the introduction section should be referenced (lines 57-65).

Checked and added refs.

Comment 5:

In line 84, Figure 3 probably means Figure 2.

Ok

Comment 6:

Sections 1 and 2 are mixed up (introduction, materials and methods and results and discussion) and then section 3 deals with results again

Maybe we have not been sufficiently clear.

Section 1 is the state of the art in the introduction. The results provided are not our results.

Section 2 is the fabrication of the samples.

Section 3 is the results after tests of memories.

Comment 7:

Some reference such as reference 1 should be written according to the journal requirements

Ok;

Reviewer 3 Report

The manuscript represents a non-volatile memory device with a crossbar structure using CNFs, where CNFs is an abbreviation of oxidized carbon nanofibers. The authors fabricated the resistance change memory device (ReRAM) using spray-gun deposition method. The manuscript is supported by the obtained results but here are some concerns to elevate the quality of the manuscript and make it representative enough for publication.

1-      A crucial proofreading is required for this manuscript o reduce the grammatical errors.

2-      Figure 3 is not reported in a proper way and it suffers from low quality.

3-      Figure 3 can be combined with figure 1 instead of a separate graph.

4-      It is suggested to move the figure 4 to the supporting information.

5-      Figure 5 is not clear enough and a high quality of the image is required.

6-      Figure 7 is not necessary, and it is better to remove this photo.

7-      The retention and endurance property are not studied, how many cycles can the device tolerate without failure?

8-      The references are not enough and there should be at least 20 citations for this manuscript.

Considering the comments, it is suggested o submit this work for another journal.

Author Response

Thank you very much for the very constructive remarks.

We tried to improve each point.

Best Regards

The manuscript represents a non-volatile memory device with a crossbar structure using CNFs, where CNFs is an abbreviation of oxidized carbon nanofibers. The authors fabricated the resistance change memory device (ReRAM) using spray-gun deposition method. The manuscript is supported by the obtained results but here are some concerns to elevate the quality of the manuscript and make it representative enough for publication.

1-         A crucial proofreading is required for this manuscript o reduce the grammatical errors.

Done

2-         Figure 3 is not reported in a proper way and it suffers from low quality.

It directly comes from the measurement system. We cannot improve it.

3-         Figure 3 can be combined with figure 1 instead of a separate graph.

Done

4-         It is suggested to move the figure 4 to the supporting information.

Sorry we do not provide supporting information in this paper.

5-         Figure 5 is not clear enough and a high quality of the image is required.

This is the best quality that we can provide.

6-         Figure 7 is not necessary, and it is better to remove this photo.

This is essential in order to show that we are using a configuration where we do not have cross-talk. Moreover, we think it is useful to show the real devices fabricated and the test phase for non-experts in the matter.

7-         The retention and endurance property are not studied, how many cycles can the device tolerate without failure?

It can endure 100 cycles (added). These are only preliminary tests.

8-         The references are not enough and there should be at least 20 citations for this manuscript.

Done.

Considering the comments, it is suggested o submit this work for another journal.

Round 2

Reviewer 1 Report

The feedback of the authors regarding my first set of comments is clear and very honest. Considering aging as the next challenge to tackle sounds very reasonable. 

I think the paper can be published as is, though I would recommend another English proof reading. 

Author Response

Paper corrected

Reviewer 2 Report

Dear authors, 

The manuscript has been significantly improved and now warrants publication in Micromachines. However some minor changes must still be addressed: 

-The verb "fabricate" in line 71 should be changed for "fabricating"

- The word "graphene" in lines 72 and 73 should be changed for "graphene oxide"

Author Response

done

Reviewer 3 Report

Majority of the reviewer's comments are not considered and further experiments are not conducted. At this state it is not acceptable and may the authors can submit to another journal. 

Author Response

What do I have to answer? what is written is not true.